# Molecular Determinants for OMF Selectivity in Tripartite RND Multidrug Efflux Systems

**DOI:** 10.3390/antibiotics11020126

**Published:** 2022-01-18

**Authors:** Esther Boyer, Jean Dessolin, Margaux Lustig, Marion Decossas, Gilles Phan, Quentin Cece, Grégory Durand, Véronique Dubois, Joris Sansen, Jean-Christophe Taveau, Isabelle Broutin, Laetitia Daury, Olivier Lambert

**Affiliations:** 1CBMN UMR 5248, Bordeaux INP, CNRS, Université de Bordeaux, 33600 Pessac, France; esther.boyer@u-bordeaux.fr (E.B.); jean.dessolin@u-bordeaux.fr (J.D.); m.decossas@cbmn.u-bordeaux.fr (M.D.); joris.sansen@u-bordeaux.fr (J.S.); jc.taveau@cbmn.u-bordeaux.fr (J.-C.T.); 2Laboratoire CiTCoM, CNRS, Université de Paris, 75006 Paris, France; m.lustig93@gmail.com (M.L.); gilles.phan@u-paris.fr (G.P.); cece.quentin@cnrs.fr (Q.C.); isabelle.broutin@u-paris.fr (I.B.); 3Unité Propre de Recherche et d’Innovation, Université d’Avignon, Equipe S2CB, 84916 Avignon, France; gregory.durand@univ-avignon.fr; 4MFP, UMR 5234, CNRS, Univiversité de Bordeaux, 33000 Bordeaux, France; veronique.dubois@u-bordeaux.fr

**Keywords:** antibiotic resistance, efflux pump, RND

## Abstract

Tripartite multidrug RND efflux systems made of an inner membrane transporter, an outer membrane factor (OMF) and a periplasmic adaptor protein (PAP) form a canal to expel drugs across Gram-negative cell wall. Structures of MexA–MexB–OprM and AcrA–AcrB–TolC, from *Pseudomonas aeruginosa* and *Escherichia coli*, respectively, depict a reduced interfacial contact between OMF and PAP, making unclear the comprehension of how OMF is recruited. Here, we show that a Q93R mutation of MexA located in the α-hairpin domain increases antibiotic resistance in the MexA_Q93R_–MexB–OprM-expressed strain. Electron microscopy single-particle analysis reveals that this mutation promotes the formation of tripartite complexes with OprM and non-cognate components OprN and TolC. Evidence indicates that MexA_Q93R_ self-assembles into a hexameric form, likely due to interprotomer interactions between paired R93 and D113 amino acids. C-terminal deletion of OprM prevents the formation of tripartite complexes when mixed with MexA and MexB components but not when replacing MexA with MexA_Q93R_. This study reveals the Q93R MexA mutation and the OprM C-terminal peptide as molecular determinants modulating the assembly process efficacy with cognate and non-cognate OMFs, even though they are outside the interfacial contact. It provides insights into how OMF selectivity operates during the formation of the tripartite complex.

## 1. Introduction

In Gram-negative bacteria, tripartite systems of the resistance nodulation cell division (RND) superfamily are multidrug efflux systems contributing to antibiotic resistance by exporting biological metabolites and antimicrobial compounds [1,2,3]. These systems are composed of an inner-membrane RND transporter driven by the proton motive force, an outer-membrane factor (OMF), and a periplasmic adaptor protein (PAP) which connects the RND transporter to OMF, therefore, forming a tripartite complex with a contiguous exit duct. The assembly of these exporting systems is an important step to achieve the functional efflux process. Deciphering the assembly mechanism is a prerequisite in the development of blockers of tripartite systems that would restore the efficiency of the existing therapeutic arsenal [4].

While PAP and RND transporters encoded by the same operon operate in pairs, the rules governing the interactions of PAP with the OMF appear less restrictive [5,6]. Indeed, different PAPs are able to bind a single OMF, e.g., TolC or OprM. In *E**scherichia*
*coli (E. coli)*, TolC can function with different couples of PAP-RND transporters but also for PAP-Major facilitator superfamily (MFS) transporters and PAP-ATP-binding cassette (ABC) transporters. In *Pseudomonas aeruginosa (P. aeruginosa)*, OprM can interact with seven of the twelve PAP-RND systems including MexA-MexB, MexC-MexD, MexE-MexF, MexX-MexY [7,8,9,10]. This versatility of interaction does not strictly apply to OMFs. One PAP can also couple more than one OMF. MexA-MexB is functional with OprM, and partially with OprJ [11,12], MexE-MexF with OprN and OprM [9], and MexX-MexY with OprM and OprA [13]. Intra- and inter-species interchangeability of components has been also observed [14,15,16]. However, this component exchange is not representative of all tripartite systems and for several other OMFs, a strict selectivity of assembly seems to operate, as for OprN that interacts only with MexE-MexF [9]. Because of this duality of selectivity and promiscuity, it remains unclear how PAPs achieve to recognize and assemble with OMFs and what are the structural determinants governing the selection of OMF by PAP.

Recent cryo-electron microscopy (cryo-EM) studies of *E. coli* AcrA–AcrB–TolC and MexA–MexB–OprM tripartite complexes have shown overall similar architectures of six PAPs surrounding one RND trimer and in a tip-to-tip interaction with the OMF, which is in an open state (Figure 1) [17,18,19]. The six periplasmic helix-turn-helix of OMF face six PAP α-hairpins, involving mainly backbone H-bond contacts. In these tripartite complexes, the OMF–PAP arrangement exhibits a reduced interfacial contact that contradicts previous biochemical and functional data [20,21,22,23,24,25,26,27], predicting a strong binding surface between the α-hairpin domain of PAP and OMF in favor of a deep-interpenetration model [28]. Interestingly gain-of-function mutants that enable non-functional chimeric efflux pumps to function have been used to identify key residues involved in the PAP–OMF assembly. Evidence of adaptative mutations far away from the tip region of the α-hairpins of AcrA, MexA (i.e., MexA_Q93R_), and *Vibrio Cholerae* VceA provide a gain of function for the chimeric AcrA–MexB–TolC, MexA–MexB–OprN, and VceA–VceB–OprM pumps [15,22,29]. Likewise, to adapt TolC to MexA–MexB, mutations that are not located at the tip of the coiled-coil domain of TolC provided a gain of function [21]. The role of these mutations which are not located in the tip-to-tip OMF–PAP contact is questioning the mechanisms of OMF recruitment in the assembly process and requires further investigations.

Here, we used the biolayer interferometry approach to investigate the interaction between several OMFs (OprN, TolC, OprM, and variant) and PAPs (MexA and MexA_Q93R_) and electron microscopy (EM) to analyze tripartite complexes in the presence of MexB. We report the reconstitution of tripartite complexes with MexA_Q93R_ and its capability to couple native OprN and TolC.

## 2. Results

### 2.1. Analysis of Mexa Binding to OMF by Biolayer Interferometry

A Q93R mutation for MexA (MexA_Q93R_) conferring a gain of function with OprN [29] is located at the α-hairpin but is not described to participate in the tip-to-tip interaction with the OMF (Figure 1).

To decipher the mechanism of action of this mutant, its interaction with various OMFs, i.e., OprM, OprN, TolC, and an OprM variant (OprM_∆473−485_) has been analyzed using the biolayer interferometry (BLI) method. Increasing concentrations of MexA_wt_ and MexA_Q93R_ variant were titrated to OMF immobilized by a biotinylated non-ionic amphipol (BNAPol) on a streptavidin biosensor and the association and dissociation were assessed by a shift in wavelength (Figure 2). Loading of BNAPol-OprM was performed under non-saturating concentrations (Appendix A).

BLI analysis revealed that k_off_ of MexA_wt_ varied depending on the OMF ligand (Table 1). The value of k_off_, being also indicative of residence time, suggested that the complex stability followed the order OprM > OprN > OprM_∆473−485_ > TolC. Unlike MexA_wt_, MexA_Q93R_ exhibited similar k_off_ values for the four OMFs suggesting that the complex stability was not dependent on the OMF. These results revealed that the OMF binding mechanisms of MexA_Q93R_ and MexA_wt_ were different.

### 2.2. Analysis of Oligomerization State of MexA_Q93R_

Previous data have shown that MexA forms a dimer in solution and a higher oligomeric state in the crystal structure [30,31,32]. The substitution of a glutamine by an arginine residue in MexA_Q93R_ introduced a charged amino acid that may affect protein–protein interactions. MexA_wt_ and MexA_Q93R_ samples were submitted to size-exclusion chromatography that showed a slight shift between elution profiles, suggesting that MexA_Q93R_ retention was reduced compared with MexA_wt_ (Figure 3A).

EM analysis of fractions corresponding to the MexA_Q93R_ peak fraction revealed complexes regular in size (Figure 3B). The average image from single-average image analysis revealed hexagonal-shaped particles with a diameter of about 8–10 nm which is compatible with a hexameric form (Figure 3B inset). EM analysis of MexA_wt_ peak fraction showed particles heterogeneous in size, reflecting the formation of aggregates when deposited on the grid (Figure 3C). This result provided evidence that MexA_Q93R_ in solution formed an oligomeric form, compatible with a hexamer.

### 2.3. Binding Analysis of MexA Variants to MexB Using BLI

Using similar conditions as performed for MexA-OMF binding analysis, various concentrations of MexA_wt_ and MexA_Q93R_ variants were titrated to MexB immobilized by BNAPol on a streptavidin biosensor (Figure 4).

BLI analysis revealed that the complex stability (k_off_ value) was slightly improved with MexA_Q93R_ compared with MexA_wt_ (Table 2). Of note, the k_off_ values were higher than that of OprM–MexA suggesting that the MexA–MexB complex was less stable than the MexA–OprM complex.

### 2.4. Impact of MexA_Q93R_ on the Formation of Tripartite Complexes

According to the BLI experiments, the Q93R mutation for MexA dramatically changed its interaction with various OMFs, we, therefore, evaluated its impact on the formation of tripartite complexes. The four OMFs (OprM, OprN, TolC, OprM_∆473−485_) and MexB stabilized in nanodiscs were mixed with MexA_wt_ or MexA_Q93R_ proteins following the method previously described [19,33]. The formation of tripartite complexes was assessed by the presence of elongated complexes observed by negative-staining EM and 2D class averaging (Figure 5 and Appendix A).

For OprN, TolC, and OprM_∆473−485_, tripartite complexes were formed with MexA_Q93R_ while no complex was observed with MexA_wt_ (Figure 5E–T). For OprM, tripartite complexes were observed with both MexA_wt_ and MexA_Q93R_ (Figure 5A–H). The overall architecture of these complexes was similar to that described previously [33]. The OprM facing the MexA–MexB complex with no direct contact between OprM and MexB was further resolved in a tip-to-tip interaction with MexA on the cryo-EM structure [19]. The formation of hybrid (non-cognate) OprN–MexA_Q93R_–MexB complexes was in good agreement with in vivo experiments reporting a gain of function with the MexA_Q93R_ variant [29]. The formation of hybrid TolC-MexA_Q93R_-MexB complexes showed that the Q93R mutation for MexA extended its interaction with TolC without the need of changing any residue at the tip-to-tip interface. Note that few atypical 2D classes of tripartite MexA_wt_–MexB–OprM complexes showed a faint contact between MexA and OprM (Figure 5C,D) probably as previously observed [33]. No such classes were encountered when tripartite complexes were generated with MexA_Q93R_ suggesting that the complexes were more stable on EM grids.

The number of tripartite complexes has been evaluated from the micrographs and reported in Table 3. The formation of a higher number of OprM–MexA_Q93R_–MexB complexes compared to OprM–MexA_wt_–MexB suggested that these tripartite complexes were assembled in a more efficient manner with MexA_Q93R_. This was also correlated by in vivo experiments where the minimal inhibitory concentration (MIC) values of ticarcillin and aztreonam for cells expressing OprM–MexA_Q93R_–MexB were twofold and fourfold higher than for those expressing native OprM–MexA_wt_–MexB (Table 4). Overall, the formation of tripartite complexes with MexA_Q93R_ was significantly improved compared to MexA_wt_, suggesting that MexA_Q93R_ had greater capabilities than MexA_wt_ to form tripartite complexes with OprM and other OMFs.

### 2.5. Impact of an OprM Variant on the Formation of Tripartite Complexes

In the assembly process of the tripartite complex, the OMF undergoes an important conformational change to achieve a tip-to-tip interaction with the PAP. The OMF switches from a closed state to an open state by the opening of its periplasmic helices [17,18,19]. Therefore, the OMF recruitment and its opening by periplasmic helices movement are two events that imply intricate interactions with PAP for which molecular details are missing.

A C-terminal-deleted mutation for OprM (OprM_∆473−485_) was used to understand by which mechanism MexA_Q93R_ promotes the assembly of tripartite complexes. OprM_∆473−485_ did not allow the production of tripartite complexes with MexA_wt_-MexB (Table 3). The inability of MexA_wt_ to form a tripartite complex with OprM_∆473−485_ correlated well with BLI experiments showing that MexA_wt_ had a higher k_off_ value for OprM_∆473−485_ than for OprM_wt,_ therefore exhibiting lower binding affinities (Table 1). These results showed that OprM 13 amino acids C-terminal peptide was of importance for MexA_wt_ binding affinity and suggested that a reduced affinity for OprM likely impaired its recruitment, and consequently, jeopardized the formation of tripartite complexes.

By replacing MexA_wt_ with MexA_Q93R_, tripartite complexes were formed with OprM_∆473−485_ meaning that MexA_Q93R_ was allowed to compensate/overcome this affinity loss due to the lack of OprM C-terminal part. However, the amount of OprM_∆473−485_–MexA_Q93R_–MexB complexes was lower than that of OprM_wt_–MexA_Q93R_–MexB (Table 3) suggesting that despite similar k_off_ values for OprM_wt_ and OprM_∆473-485_, MexA_Q93R_ did not permit to fully compensate the lack of C-terminal part of OprM (Table 1). It seemed that MexA_Q93R_ was acting more on stabilizing an OMF–PAP complex than on the recruitment step of OMF that was yet in good accordance with the BLI experiment, showing similar k_off_ values of MexA_Q93R_ for the four OMFs.

### 2.6. The Increase in Antibiotic Resistance Is Related to a Q93R Mutation When Associated with D113 Residue

In the cryo-EM tripartite complexes, OprM interacts with six MexA molecules, the six α-hairpins of MexA forming a tight helical bundle (Figure 6A). By substituting Q93 neutral residue with R93 charged residue, the latter is closer to the adjacent D113 residue and the distance between side chains (2.96 Å) is compatible with an ionic bond (Figure 6B). Energies associated with the formation of the hexamer of MexA alone have been estimated with SymmDock. Molecular docking predicted hexameric MexA complexes with an energy score in favor of MexA_Q93R_ indicating a better stabilization of the MexA_Q93R_ complex (Appendix A). This hexamer was assembled in a tip-to-tip interaction with OprM using PatchDock (Figure 6B). The formation of an interchain electrostatic interaction between D113 and R93 residues provided a clue on how the introduction of an arginine residue contributes to stabilizing a hexameric structure of MexA_Q93R_.

To assess that the residue D113 would act in synergy with R93, an antibiotic susceptibility assay was performed. For that, the *P. aeruginosa* PAO1 strain was transformed with plasmids carrying genes encoding OprM, MexB, and MexA variants. The strain transformed with MexA–MexB–OprM is two-fold more resistant than native PAO1 which could be due to a slight increase in the level of expressed MexA–MexB–OprM system (Table 4). The introduction of the Q93R mutation in MexA resulted in a two-fold increase in the resistance of the complemented strain to ticarcillin and aztreonam. In order to evaluate the importance of the potential hydrogen bond formed between MexA-R93 and MexA-D113 (Figure 6B), the latter was mutated in alanine. Strains that expressed MexA_D113A_–MexB–OprM or MexA_D113A + Q93R_–MexB–OprM showed that the MICs of ticarcillin and aztreonam were two times lower than strains expressing MexA–MexB–OprM (Table 4). This result provided evidence that the pair residues D113 combined with R93 are involved in the increase in antibiotic resistance.

## 3. Discussion

RND efflux transporters are functional when assembled in tripartite complexes with PAP and OMF partners. Deciphering how they achieve assembly is of importance for medical treatment due to the contribution of these complexes in both multidrug resistance and virulence. With recent advances in elucidating the structure of tripartite assemblies, the OMF and PAP are coupled together via a limited protein–protein interface (so-called tip-to-tip interaction), that still does not permit untangling the tricky knots of OMF selectivity [5,6].

We show that the formation of tripartite complexes coupling OprN, TolC, and OprM_∆473−485_ can be achieved with a MexA variant (MexA_Q93R_) while it was not successful with MexA_wt_. The mutated residue is located at the α-hairpin but too far for interacting directly with the OMF. Although this Q93R mutation for MexA did not originate from a pathogenic strain and presents poor clinical importance, it has been selected as a gain-of-function mutant and provides a clue for understanding the assembly process of RND tripartite systems. Indeed, it points out that putative paired anionic and cationic residues (R93, D113) between two adjacent protomers could stabilize the hexameric structure of MexA_Q93R_. A comparative analysis of the amino acid sequences of other PAPs showed that similar couples of residues are present for native PAPs. MexX possesses a putative couple of residues (K102–E122) located at the same position as R93–D113 for MexA (Appendix A). In the absence of a MexX structure, a model has been predicted using the I-TASSER server [34,35,36] and a C6 hexamer model built with SymmDock [37]. The charged groups of K102 and E122 are at a reasonable proximity to establish electrostatic interactions suggesting that it could be used as an asset for MexX-MexY when forming a tripartite complex with OprM or/and with OprA (Figure 6C). In the MexE sequence, residues R97 and E128 are located in the α-hairpin and could form favorable electrostatic interactions between paired anionic cationic side chains (Figure 6D). Like for MexX, these interprotomer interactions mediated by these two residues may help in the formation of MexE–MexF–OprN or/and MexE–MexF–OprM complexes.

This analysis of tripartite system assembly highlights important molecular determinants for PAP–OMF interaction that are not directly involved in the tip-to-tip interaction. As OMF determinants, we have identified that the C-terminal part was of importance for forming tripartite complexes. The implication of the C-terminal part has been previously reported for functional OprM–MexA–MexB [15,38,39] and TolC–AcrA–AcrB [40,41]. Our results indicate that the deletion of 13 amino acids of the C-terminal end of OprM has a dramatic effect on the formation of tripartite OprM–MexA–MexB complexes. BLI experiments showed that MexA has a reduced affinity for OprM_∆473−485_ suggesting that the efficacy of tripartite formation relies on the presence of this C-terminal part. This C-terminal part originates from the equatorial domain but its complete structure has not been solved in both crystal and cryo-EM structures, probably because of high flexibility. It is unlikely that its role in the assembly process occurs at the stage of the tip-to-tip interaction (too short in length) but it might participate directly or indirectly in a transient interaction with MexA, that would occur earlier than the stable tip-to-tip interaction. This transient interaction may help in OprM recruitment by MexA and altering the binding affinity of MexA for OprM decreases the efficacy of tripartite complex formation. Our results did not provide details on the protein interfaces involved in this step. However, biochemical and functional data previously suggested lateral contacts between α-hairpin of PAP and OMF helices and could well fit in an enlarged assembly sequence with transient interactions preceding the tip-to-tip contact.

As a MexA determinant, the Q93R mutation successfully produced tripartite complexes with cognate and non-cognate OMFs. Interestingly, bacteria were less susceptible to antibiotics with MexA_Q93R_ than with MexA_wt_, and the amount of tripartite complexes was increased. This mutation promotes the hexameric organization of MexA mediated by a putative interprotomer ionic bond (Figure 6). During the assembly process, this mutation likely promotes or stabilizes the formation of the six-helix bundle of MexA contacting OprM, which may trigger OprM opening and/or stabilize the tip-to-tip contacts. Improving the efficiency of the opening/stabilization of OMF-PAP in a tip-to-tip contact likely allows compensating for the lack of the C-terminal part for OprM_∆473−485_ needed for the previous transient interaction described above. This hypothesis is in good accordance with the previous study on VceA–VceB–OprM complex assembly, reporting on the role of the C-terminal domain of OprM and VceA α-hairpin [15]. In addition, this Q93R mutation extends the capability of MexA to assemble with non-cognate OprN and TolC partners. According to protein–protein docking, they are predicted to interact with a lower energy binding (Appendix A). The PAP–OMF interface also imposes an OMF selectivity that can be overcome by reinforcing PAP self-assembly capability.

## 4. Materials and Methods

### 4.1. Material and Reagents

1-palmitoyl-2-oleoyl-*sn*-glycero-3-phosphocholine (POPC) was purchased from Avanti Polar Lipids (Alabaster, AL, USA). Sodium cholate hydrate, octyl-β-d-glucopyranoside (OG), and *n*-Dodecyl β-d-maltoside (DDM) were purchased from Sigma-Aldrich (St Louis, MI, USA). SM2 Bio-Beads were obtained from Bio-Rad (Hercules, CA, USA). Superdex 200 PC 3.2/30 and Superose 6 Increase 3.2/300 columns were purchased from Cytivia (Freiburg, Germany). EM grids (Cu 300 mesh) were purchased from Agar Scientific (Stansted, UK). High precision streptavidin biosensors (SAX) for BLI analysis were purchased from Sartorius (Göttingen, Germany).

### 4.2. Protein Preparation

Two membrane scaffold proteins, MSP1D1 and MSP1E3D1 (genetic constructs available from AddGene, Cambridge, MA, USA) expressed and purified from bacteria, were used to make OMFs and MexB nanodiscs respectively. Proteins (MexB, MexA_wt_ and MexA_Q93R_, OprM and OprM_∆473−485_, OprN and TolC) were expressed and purified from bacteria as previously described [33,42]. After purification, protein buffers contained 1.5% OG for TolC, 0.03% DDM for MexB and 0.05% DDM for MexA, OprN, and OprM.

### 4.3. Membrane Protein Stabilization with Amphipols

BNAPols (biotinylated non-ionic amphipols) were synthesized by free radical telomerization of an amphiphilic monomer, carrying two glucose moieties and a single undecyl alkyl chain, in the presence of a thiol-based transfer agent bearing a single azido group. The biotin function was subsequently connected to the polymer through a Huisgen cycloaddition as previously described [43]. The BNAPol used in the study had an average molecular mass of ~14.9 kDa and a number-average degree of polymerization of ~20. The extent of grafting of the biotin group was estimated to be ~40% per polymer chain. The membrane protein solution was mixed with BNAPol solution at a 2:1 BNAPol:membrane protein mass ratio for 2 h at 4 °C in a 10 mM Tris/HCl, pH 7.4, 100 mM NaCl 0.01% NaN_3_, and 0.02% DDM buffer. Detergent was removed by the addition of SM2 Bio-beads with gentle shaking for 3 h at 4 °C. After centrifugation, the mixture was subjected to size-exclusion chromatography (Superdex 200 PC 3.2/30) equilibrated with 10 mM Tris/HCl, pH 7.4, 100 mM, NaCl 0.01% NaN_3_ buffer at 0.05 mL min^−1^.

### 4.4. Binding Analysis Using BLI

Each binding assay was performed with BLItz™ device (ForteBio Inc., Fremont, CA, USA) at room temperature in 10 mM Tris/HCl, pH 7.4 100 mM NaCl 0.01% NaN_3,_ and 0.05% DDM buffer. OMFs and MexB, stabilized into BNAPols, were immobilized on SAX biosensors and exposed to a range of MexA concentrations from 0 to 200 µM. BLItz Pro™ software (version 1.2.1.5, ForteBio Inc. Fremont, CA, USA) was used to fit the curves and to process the data.

### 4.5. Formation of Tripartite Complexes

POPC lipids were dissolved in chloroform, then dried under vacuum using a rotatory evaporator. The lipid film was suspended in the reconstitution buffer (10 mM Tris/HCl, pH 7.4, 100 mM NaCl) and subjected to 6 rounds of 5′ sonication at 5 watts. Lipid concentration was quantified by phosphate analysis [44].

Tripartite complexes were assembled according to the protocol previously described [33] with slight modifications. Briefly, insertion of OMFs (i.e., OprM, OprN, TolC) in nanodiscs and MexB in nanodiscs using MSP1D1 and MSP1E3D1, respectively, was performed as follows. OMF and MexB solutions were mixed with POPC liposomes and MSP solution at a final lipid/MSP/protein molar ratio of 23:1:0.6 for OMFs (except for TolC, 31:1:2.4) and 32:1:0.5 for MexB in a 10 mM Tris/HCl, pH 7.4, 100 mM NaCl and 15 mM Na-cholate solution. Detergent was removed by the addition of SM2 Bio-Beads into the mixture shaken overnight at 4 °C. Tripartite complexes were assembled in vitro by mixing the OMF and MexB solution with MexA_wt_ or MexA_Q93R_ solution, at a MexA:MexB:OMF ratio of 10:1:1 in 10 mM Tris/HCl, pH 7.4, 100 mM NaCl 0.01% NaN_3_ and 0.02% DDM buffer. Mixtures were incubated at 20 °C shaking for 7 days before EM grid preparation.

### 4.6. Analysis of MexA Oligomerisation State

MexA_wt_ and MexA_Q93R_ in purification buffer were subjected to size-exclusion chromatography on a Superdex 200 PC 3.2/30 column, equilibrated with 10 mM Tris/HCl, pH 7.4, 100 mM NaCl 0.01% NaN_3_ and 0.05% DDM buffer at 0.05 mL min^−1^.

### 4.7. Electron Microscopy Acquisition and Image Analysis

For EM grid preparation, a diluted mixture of the sample suspension was deposited on a glow-discharged carbon-coated copper 300 mesh grids and stained with 2% uranyl acetate (*w*/*v*) solution. Images were acquired on a Tecnai F20 electron microscope (ThermoFisher Scientific, Waltham, MA, USA)) operated at 200 kV using an Eagle 4k_4k camera (ThermoFisher Scientific). Image alignment and two-dimensional averages were performed with Eman2 [45] using a reference-free alignment procedure. For MexA_Q93R_, MexA-MexB-OprM, MexA_Q93R_-MexB-OprM, MexA_Q93R_-MexB-TolC, and MexA-MexB-OprM_∆473−485_, a total of 11,572, 19,260, 46,145, 1191, and 14,025 particles, respectively, were automatically selected and processed for class averaging. For MexA_Q93R_-MexB-OprN, 1236 particles were manually selected and processed like the others. For assessing the occurrence of tripartite complexes, 150 micrographs were randomly collected per grid. The number of complexes was estimated by manual picking on a set of 50 micrographs. The experiment was conducted in triplicate and expressed as the mean and standard error of the mean (sem).

### 4.8. Model Simulation and Score Evaluation

The SymmDock server [37,46] was used to produce C6 hexamer MexA_wt_ (PDB: 6TA5) and MexA_Q93R_ after mutating Q93 to R93 with Discovery Studio Visualizer (BIOVIA, San Diego, CA, USA). MexA_Q93A_ and MexA_D113A_ hexamers were generated using the same procedure. The PatchDock server [37] was used to simulate MexA hexamer-OMF trimer assembly, with fully rigid multimers. The FireDock algorithm allowed a refinement of the obtained complexes and estimated the binding energy (Appendix A). During this refinement, the previous complex is modified in order to enhance the interface between the proteins. OprN (PDB: 5IUY) was modeled in an open state with Modeller [47], based on OprM (6TA5 chain A). OprM, modeled OprN, and TolC (PDB: 5NG5) were symmetrized with SymmDock before being submitted to PatchDock. MexX and MexE monomeric chains were obtained from the I-TASSER server and submitted to SymmDock to generate a hexameric form. Examination of the proximity between pairs of residues in adjacent chains was examined and K102 and E122 in MexX and R97 and E128 in MexE presented possible interactions.

### 4.9. Measurement of Antibiotic Susceptibility

The complete coding sequence corresponding to the operon *mexA-mexB-oprM* from *P. aeruginosa* PAO1 (472024–477790) (Accession No. GCF_000006765.1) was amplified by high-fidelity PCR and cloned into the pUCP24 plasmid by assembly. Then, specific mutations (D113A, Q93R, and D113A + Q93R) were inserted by site-directed mutagenesis following the recommendations of the supplier (New England Biolabs France, Evry, France). Recombinant plasmids were transferred into *E. coli*-competent cells (DH10B) by heat shock and cultured at 30 °C to avoid unspecific recombination. The sequence of the cloned and mutated *mexA-mexB-oprM* was verified by Sanger sequencing. Recombinant plasmids were then transferred into the PAO1 strain by electroporation. The recombinant strains were selected on MH medium supplemented with 10 μg/mL gentamicine. The mutated plasmid-borne efflux system was compared with the wild-type plasmid-borne one to assess the impact of the mutations. MICs to ticarcillin and aztreonam were performed following CLSI recommendations.

## 5. Conclusions

In conclusion, we provide evidence that the OMF selectivity does not rely only on molecular determinants of the tip-to-tip OMF–PAP interface described in the tripartite complexes, but also on additional molecular determinants on PAP and OMF that allosterically modulate the formation of tripartite complexes. Further investigations are needed to fully elucidate the molecular mechanisms underlying the formation of RND tripartite complexes.

## Figures and Tables

**Figure 1 antibiotics-11-00126-f001:**
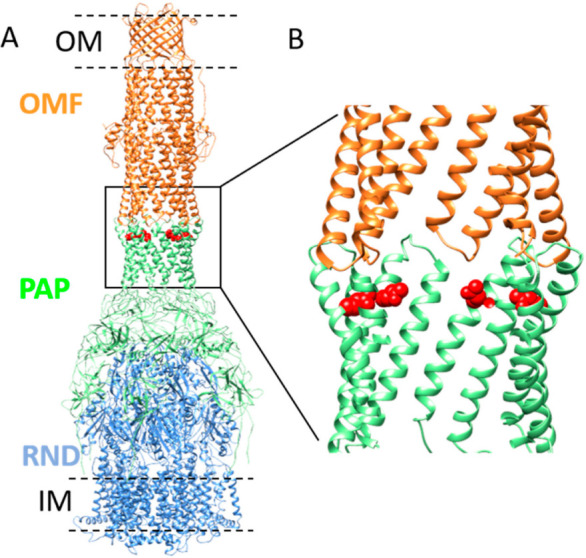
Model of MexA–MexB–OprM tripartite complex and position of Q93 residue in α-hairpin of MexA. (**A**) Model representation of OprM–MexA–MexB tripartite complex (PDB: 6TA5) showing OprM (OMF component) trimer (colored in orange) and MexB (RND component) trimer (colored in blue) connected by MexA (PAP component) hexamer (colored in green). The outer membrane (OM) and inner membrane (IM) are schematically drawn (black dashed lines). The position of the residue Q93 is shown in red (side chain). The position of V472 (or V455 in mature OprM sequence numbering) corresponding to the C-terminal residue solved in OprM structure is indicated on two protomers in the equatorial domain (black arrows). Residues T473–A485 are not visible in the structure. (**B**) Close-up view of the position of the Q93 residue relative to the tip-to-tip contact between OprM and MexA.

**Figure 2 antibiotics-11-00126-f002:**
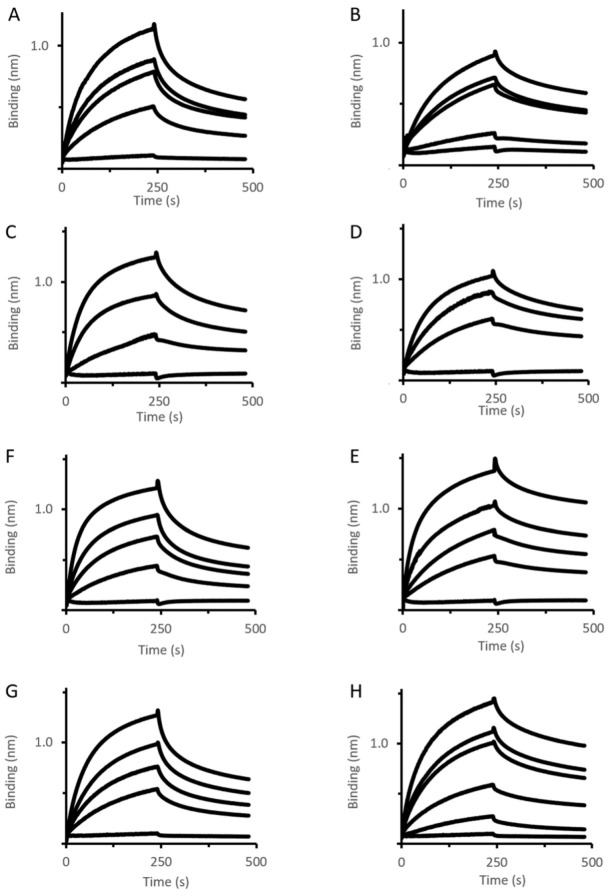
OMF–PAP interactions assessed by BLI. Immobilized BNAPol-OprM (**A**,**B**), BNAPol-OprN (**C**,**D**), BNAPol-TolC (**E**,**F**), BNAPol-OprM_∆473−485_ (**G**,**H**) were exposed to different concentrations (from 0 to 100 µM) of MexA_wt_ (left column) or MexA_Q93R_ (right column). Interactions (association and dissociation) were assessed by a wavelength shift (nm). All reactions were performed at room temperature.

**Figure 3 antibiotics-11-00126-f003:**
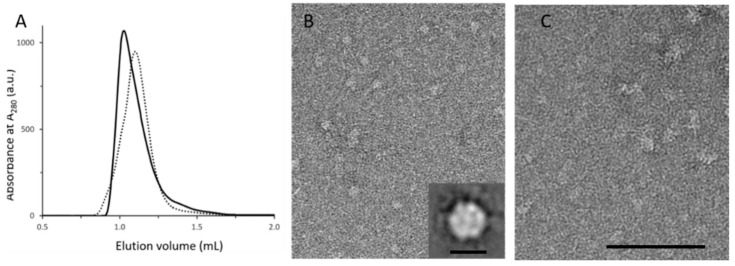
Analytical characterization and EM analysis of MexAQ93R and MexAwt. (**A**) Analytical size-exclusion chromatography (SEC) analysis of MexAQ93R (solid trace) and MexAwt (dotted trace) samples. (**B**) EM analysis of the SEC peak fraction of MexAQ93R exhibiting circular particles. Inset: average image showing a hexagonal-shaped particle with a diameter of about 8–10 nm. Scale bar 10 nm. (**C**) EM analysis of the SEC peak fraction of MexAwt showing heterogenous particles in size compared with (**B**). Scale bar 100 nm.

**Figure 4 antibiotics-11-00126-f004:**
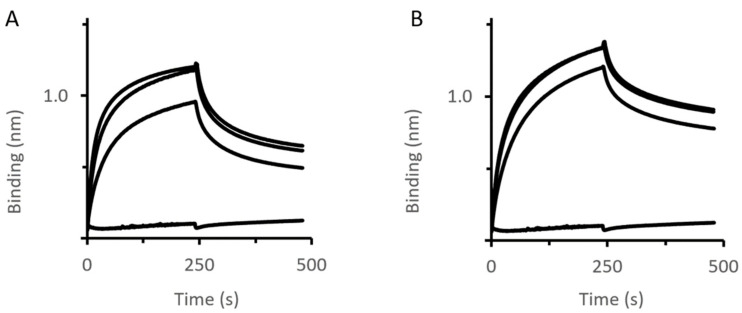
MexB–PAP interactions assessed by BLI. Immobilized BNAPol-MexB were exposed to different concentrations (from 0 to 200 µM) of MexA_wt_ (**A**) or MexA_Q93R_ (**B**). Interactions (association and dissociation) were assessed by a wavelength shift (nm). All reactions were performed at room temperature.

**Figure 5 antibiotics-11-00126-f005:**
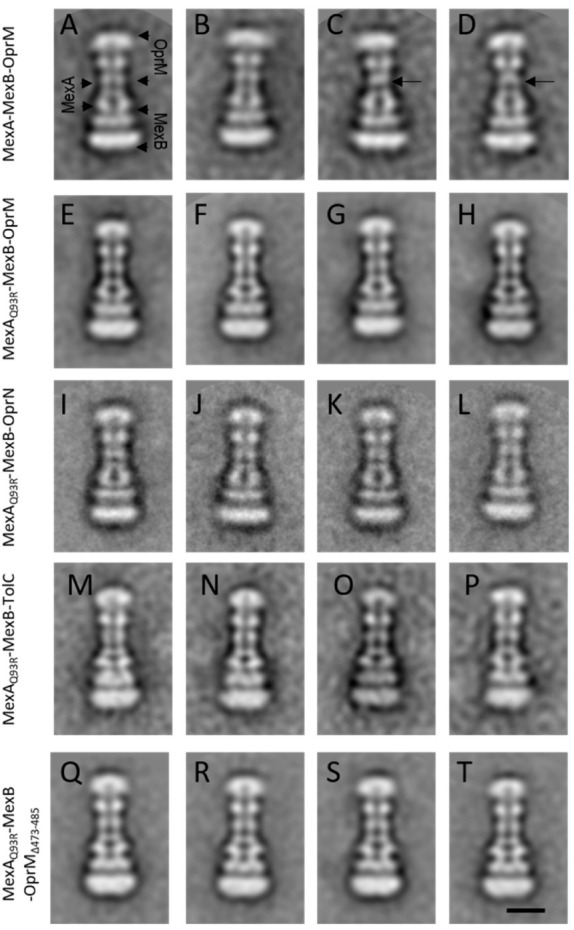
Single-particle analysis of tripartite complexes. Representative 2D classes of tripartite complexes MexA–MexB–OprM and derivatives observed by negative-staining EM. (**A**–**D**) MexA–MexB–OprM complexes. Typical classes (**A**,**B**) showing a continuous canal between OprM and MexA. Atypical classes (**C**,**D**) exhibiting a faint contact between OprM and MexA (back arrows). (**E**–**H**) MexA_Q93R_–MexB–OprM complexes. (**I**–**L**) MexA_Q93R_–MexB–OprN complexes. (**M**–**P**) MexA_Q93R_–MexB–TolC complexes. (**Q**–**T**) MexA_Q93R_–MexB–OprM_∆473−485_ complexes. Note that when formed with MexA_Q93R_, tripartite complexes exhibited an open coupled OMF whatever the considered class, unlike MexA_wt_ for which several classes presented closed coupled OMF. Scale bar 10 nm.

**Figure 6 antibiotics-11-00126-f006:**
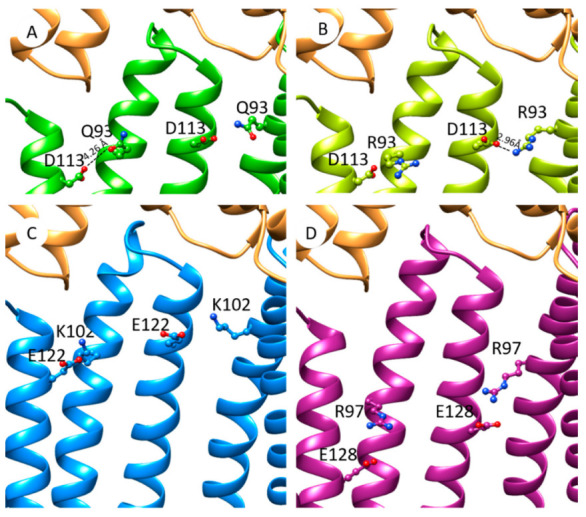
Hypothetic model of PAP interprotomer stabilization mediated by paired anionic-cationic residues in the tip-to-tip PAP–OprM contact. (**A**) Focus on OprM–MexA contact from cryo-EM model of OprM–MexA–MexB (PDB: 6TA5). Positions of residues Q93 and D113 are shown on MexA (colored in green). OprM is colored in orange. (**B**) Model of MexA_Q93R_ (light green) interacting with OprM (orange). Distance between R93 and D113 (2.96 Å) is compatible with interprotomer electrostatic interactions. (**C**) Model of MexX (blue) interacting with OprM with putative interprotomer interactions mediated by side chains of K102–E122 residues. (**D**) Model of MexE (purple) interacting with OprM with putative interprotomer interactions mediated by R97–E128 side chains.

**Table 1 antibiotics-11-00126-t001:** Kinetics parameters for the OMF–PAP interaction using biolayer interferometry.

Ligand	Analyte	k_off_ (10^−3^ s^−1^)	k_on_ (10^2^ M^−1^s^−1^)	K_D_ (µM)
OprM_wt_	MexA_wt_	2.15	1.80	12.0
OprM_∆473−485_	MexA_wt_	4.58	1.03	44.0
OprN	MexA_wt_	3.58	1.77	20.0
TolC	MexA_wt_	5.8	1.27	45.8
OprM_wt_	MexA_Q93R_	2.66	0.81	32.9
OprM_∆473−485_	MexA_Q93R_	2.63	1.02	25.7
OprN	MexA_Q93R_	2.38	0.88	26.9
TolC	MexA_Q93R_	1.91	1.08	17.8

Data fitting using Langmuir 1:1 model.

**Table 2 antibiotics-11-00126-t002:** Kinetics parameters for the OMF–PAP interaction using biolayer interferometry.

Ligand	Analyte	k_off_ (10^−3^ s^−1^)	k_on_ (10^2^ M^−1^s^−1^)	K_D_ (µM)
MexB	MexA_wt_	5.5	2.50	23.0
MexB	MexA_Q93R_	3.0	1.73	17.4

Data fitting using Langmuir 1:1 model.

**Table 3 antibiotics-11-00126-t003:** Estimation of tripartite complexes amount from electron microscopy fields.

	PAP	
OMF	MexAwt	MexA_Q93R_
OprM_wt_	1146 ± 59	1981 ± 156 *^a^
OprM_∆473−485_	0	589 ± 15 *^b,^ **^c^
OprN	0	10 ± 0.3 **^b^
TolC	0	164 ± 3 **^b^

Complexes were counted from 3 sets of 50 micrographs. Data are the mean ± sem. Student’s test significantly different (* 0.01 < *p* < 0.05; ** 0.001 < *p* < 0.01). ^a^ Compares MexA–MexB–OprM with MexA_Q93R_–MexB–OprM; ^b^ compares MexA_Q93R_–MexB–OMF with MexA_Q93R_–MexB–OprM; ^c^ compares MexA_wt_–MexB–OprM with MexA_Q93R_–MexB–OprM_∆473−485._

**Table 4 antibiotics-11-00126-t004:** Antimicrobial susceptibility of cells expressing MexA variants.

	Minimal Inhibitory Concentration (MIC, µg/mL)	
Strain	Ticarcillin	Aztreonam
PAO1	32	4
PAO1 pUCP24-*mexAB-oprM* *wt*	64	8
PAO1 pUCP24-*mexA _D113A_ mexB-oprM*	32	4
PAO1 pUCP24-*mexA* *_Q93R_ mexB-oprM*	128	32
PAO1 pUCP24-*mexA* *_D113A + Q93R_ mexB-oprM*	32	4

## Data Availability

The data presented in this study are available on request from the corresponding authors.

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
