# Peer review of "Molecular Determinants for OMF Selectivity in Tripartite RND Multidrug Efflux Systems"

_antibiotics, 2022, doi:10.3390/antibiotics11020126_

Round 1
Reviewer 1 Report
In "The Structural and Functional Study of Efflux Pumps Belonging to the RND Transporters Family from Gram-Negative Bacteria", Boyer and colleagues show that the Q93R mutation of MexA increases antibiotic resistance against ticarcillin and aztreonam. Their analyses indicate that this mutation would allow the formation of an ionic bond between R93 and D113 in MexA (an interpromoter interaction that contribute to stabilize the hexameric structure of MexAQ93R). These results would explaine how this mutation can affect the function of the efflux pump despite being far away from the tip region of the α-hairpin of MexA.
The study provides interesting information about the effects of MexAQ93R mutation. The manuscript is clear and easy to read. The methods applied and data interpretation are appropiate.
As comment, it is not completely clear to me how the antimicrobial susceptibility test was performed. In the study, the authors cloned the corresponding variants of the mexAB-oprM system into the pUCP24 vector and then transformed the P. aeruginosa PAO1 strain with them. But this strain has the wild type copy of the mexAB-oprM system in the genome, so for all the experiments they had a mixture of plasmid-encode and genome-encode transporters.
Also, the desired transporter was cloned into a vector, but the antibiotic used to mantain the plasmid in the strain during the susceptibility tests was not described. Did the authors use any antibiotic? An explanation/justification for these points would be helpful.
Also, I would suggest a few modifications to improve the final version of the manuscript:
-Check figure 2 format: there are some black boxes (next to letter C and F) that are not visible in the pdf file, but that can be seen by other programs (or when the figure is printed).
-In figure 4 there are some very thin lines in parts of the graphs (for example, in the upper right part of the graph in 4B, and above of the 250 s mark in graph 4A).
-At line 117, replace "oligomerisation" with "oligomerization".
-In figure 5, I would recommend to add the corresponding label to each row (such as "MexA-MexB-OprM" for A-D) in the left part of the figure, since it would be easier to read.
Altogether, the article provides a clear and compelling story about MexAQ93R mutation, and I hope it is published after minor modifications.
Author Response
Based on the recommendation of the reviewer, we have prepared a revised version of our manuscript taking into account all points raised.
* Regarding the antimicrobial susceptibility test:
Since MexAB-OprM participates to the natural resistance of Pseudomonas aeruginosa, any change of its functional units would decrease its susceptibility to these molecules. Thus, we compared the mutated plasmid-borne efflux system with the wt plasmid-borne one to deduce the impact of these mutations. The expression of the plasmid-borne transporters was controlled with the pLac promoter present on the plasmid. Moreover, the MICs were performed in presence of gentamicin. We add the following sentence:
The recombinant strains were selected on MH medium supplemented with 10 μg/mL gentamicine. The mutated plasmid-borne efflux system was compared with the wild-type plasmid-borne one to assess the impact of the mutations.
* Regarding figure 2 and 4 we use a different format. We hope that these black boxes are no longer present. Figure 5 has been modified according reviewer’s suggestion.
Reviewer 2 Report
The manuscript 'Molecular determinants for OMF selectivity in tripartite RND multidrug efflux systems' is a well-written, interesting and well-organized work that studies the assembling of the tripartite RND-PAP-OMF efflux pumps used by resistant bacteria to expel out of them the antibiotics, preventing them to exert their antibacterial action. In addition, it is studied how specific mutations interfere with this assembling, finding a mutation that favours it in specific constructs, and not only that: it also increases the resistence of the bacteria to specific antiobiotics.
The work is outstanding as it explores a crucial mechanism in bacterial resistance and improves the knowledge of the processes involved in this resistance mechanism. It has an elegant design, a good experimental execution and it is adequately referenced. Introduction is very good also, with a figure it would be perfect. Therefore, this work can be published in Antibiotics after a few modifications that will improve it, given below:
Concerns
1) Please define all abbreviations, including the ones that refer to genes/proteins names, after first appearance.
2) Consider adding in the introduction a schematic figure to illustrate how the RND-PAP-OMF is assembled in the membrane, it would improve a lot the introduction (which is excellent).
3) Add cities and countries to each provider mentioned in Material and Methods. The majority are not situated (AddGene, Bio-Rad, Sartorius, ForteBio, ThermoFisher...). Not only in the Materials and reagents subsection, also for the equipments used and mentioned in the following subsections.
4) Please check this sentence "13 amino acids C-terminal end". Should it be "13 amino acids of the C-terminal end"? (line 301)
Author Response
Based on the recommendation of the reviewer, we have prepared a revised version of our manuscript taking into account all points raised.
* Abbreviations of proteins and genes are now explained.
* Regarding the schematic figure in the introduction, Figure 1 is slightly modified with annotations that help the reader and is quoted in the paragraph describing the assembly (line 60);
* The provider addresses ( Cities and countries) are now mentioned.
* In the sentence (line 301), change has done accordingly “13 amino acids of the C‑terminal end”